JGURD: joint gradient update relational direction-enhanced method for knowledge graph completion

Ding Lianhong 1
Li Mengxiao 1
Gao Shengchang 2
Li Juntao 1
Yuan Ruiping 1
Yu Jianye 1 yujianye@bwu.edu.cn
1 School of Information, Beijing Wuzi University , Beijing , China
2 Information Department, People’s Hospital Affiliated to Shandong First Medical University , Shandong , China
Angiulli Giovanni
Electronic publication date: 2025 Apr 18
Publication date: 2025
Volume: 11
Electronic Location ID: e2808
Received 2024 Jul 2; Accepted 2025 Mar 16
Copyright: © 2025 Ding et al.
Copyright year: 2025
Copyright holder: Ding et al.
License: This is an open access article distributed under the terms of the Creative Commons Attribution License, which permits unrestricted use, distribution, reproduction and adaptation in any medium and for any purpose provided that it is properly attributed. For attribution, the original author(s), title, publication source (PeerJ Computer Science) and either DOI or URL of the article must be cited.
License URL: https://creativecommons.org/licenses/by/4.0/

Keywords: Knowledge graph completion, Graph neural networks, Joint gradient update, Link prediction, Relational direction, Encoder-decoder, Multi-relational graph

Funding: National Key R&D Program of China 2023YFC3805703 National Science and Technology Major Project of China J2019-VI-0004-0117 National Natural Science Foundation of China 72101033 and 62172393 Beijing Municipal Education Commission KZ202210037046 Excellent Science and Technology Innovation Team Project in Tongzhou District CXTD2023010 This work was supported by the National Key R&D Program of China (No. 2023YFC3805703), National Science and Technology Major Project of China (J2019-VI-0004-0117), National Natural Science Foundation of China (72101033, 62172393), Key Project of Science and Technology Plan of Beijing Municipal Education Commission (KZ202210037046), Excellent Science and Technology Innovation Team Project in Tongzhou District (CXTD2023010). The funders had no role in study design, data collection and analysis, decision to publish, or preparation of the manuscript.

==============================
Relational direction plays an important role in multi-relational knowledge graphs (KGs). Current knowledge graph completion (KGC) methods suffer from insufficient utilization of relation correlation information. To address this issue, this article proposes a novel KGC framework, namely JGURD, which uses the encoder-decoder structure to achieve Joint Gradient Update with Relational Direction information. It combines graph convolutional networks (GCNs) with KG embedding methods, defining a update mechanism for entities and relationships to joint gradient updates. To incorporate entity information into the update of relationships, the forward propagation gradients of the triple score function are recorded, and entity gradient information is fused into relationship updates. To fully utilize relational direction information, a relation correlation graph (RCG) is constructed based on the topological patterns of relationship pairs. We design a multi-relation encoder combining GCN and multi-layer attention mechanism on RCG to comprehensively capture local and global structures of the RCG. To enhance the interpretability and adaptability of JGURD, three different decoders are employed. Experimental results show that JGURD outperforms the second-place HHAN-KGC, and the Hits@3 and MRR metrics on the FB15k dataset increased by 6.8% and 8.9%, respectively.

Introduction

In recent years, large-scale knowledge bases such as Freebase (Bollacker et al., 2008), DBpedia (Lehmann et al., 2015), and YAGO (Lehmann et al., 2015), have been developed to store and organize structured public knowledge. These knowledge bases are abstracted as directed multi-relational graphs, forming knowledge graphs (KGs) (Peng et al., 2023). In KGs, numerous facts are constructed in the form of triples, consisting of head entity, relationship, and tail entity, denoted as (u,r,v). For example, the triple (James Cameron, directs, Avatar) represents the fact that “James Cameron is the director of the movie Avatar”.

KGs are often constructed semi-automatically or manually, which can result in inaccurate and incomplete entity information. To address this issue, knowledge graph completion (KGC) (Chen et al., 2020) have emerged. It predicts the missing parts of a triple using known entities, relationships, attributes, or other information. As shown in Fig. 1, the dashed line indicates the predicted relationship between “Angel di Maria” and “Football Player.” This relationship can be inferred from existing triples, forming a new factual triple: (“Angel di Maria,” “isA,” “Football Player”).

Figure 1 A subgraph of the KG about football team.

Most existing methods adopt an independent update strategy, focusing solely on feature aggregation for entities, while relationship updates are independent of entities and fail to consider the dynamic changes of entities (Vashishth et al., 2020; Cai et al., 2019; Ye et al., 2019). This strategy may result in relationship representations being unable to effectively capture the interaction information between entities, thereby weakening the role of relationship representations in capturing semantic associations and supporting entity representation learning. Furthermore, if the impacts of incoming and outgoing relationships are not explicitly distinguished when processing neighboring node features, it may lead to the loss or confusion of relational direction information. For the example illustrated by Fig. 1, in the triples (Argentina, Have, Forward) and (Lionel Messi, Belongs to, Argentina), “Argentina” serves as the central entity with both outing and incoming relationships. If the features of neighboring nodes are simply aggregated with weights, the representation of “Argentina” may conflate the semantic sources of “Forward” and “Lionel Messi,” making it difficult to distinguish the distinct contributions of incoming and outgoing relationships. KGs with a large number of relationships, where interactions between entities and relationships become increasingly sparse and complex as graph scales, and where the directionality of relationships needs to be carefully considered, are more likely to exhibit the aforementioned issues.

Through the analysis of the above example, we can identify the challenges that KGC may face. These include insufficient dynamic interaction between entity and relationship representations due to the independent update strategy in existing methods, semantic ambiguity from the loss of relational directional information, and limitations in feature expression caused by the lack of directional distinction during neighborhood aggregation. To address these challenges, this article proposes a novel KGC method, JGURD, which integrates relational direction information within a joint gradient update framework. Specifically, this article builds on the TACT (Chen et al., 2021) approach to divide relationship pairs into six topological structures, constructing a relation correlation graph (RCG) to capture multi-relational topological patterns. Based on this, a multi-relation encoder is proposed, leveraging graph neural networks (GNNs) to extract local and global structural information of relationships. Additionally, a relation direction-aware entity encoder is introduced to enhance entity embeddings by integrating relational information from different directions. Finally, a joint gradient update mechanism is proposed to dynamically optimize entity and relationship representations, ensuring their interdependence and dynamic interaction during the optimization process.

The main contribution of this article is as follows: 1) We design a multi-relation and entity encoding mechanism. For multi-relation KGs, we construct a RCG that integrates local structure information and global dependencies to capture the topological interactions and semantic features between relationships. Additionally, a relation direction-aware entity encoder is introduced to enhance entity representations by incorporating relational direction information.

2) We propose a joint gradient update mechanism that records the forward propagation gradients of entities and relationships, and iteratively updates them using a message-passing process, enabling dynamic interaction and collaboration between entity and relationship representations during optimization.

3) We employ the scoring functions of TransE, RotatE, and DistMult as decoders and conduct experiments on multiple benchmark datasets to validate the effectiveness of the proposed method.

Related work

In recent years, researchers have been continuously exploring new methods to enhance the performance of KGC models, including distance translation based models, semantic matching based models and neural network models.

Distance translation-based models map entities and relationships to low-dimensional vector space, representing a relationship through the translation of distance to reflect the semantic association between entities. Translating embeddings for modeling multi-relational data (TransE) (Bordes et al., 2013) minimizes the distance between the sum of the head entity vector and the relationship vector, and the tail entity vector. However, it can only effectively handle simple 1-1 relationships and fails to address more complex types, such as reflexive, 1-N, N-1, and N-N relationships. Building on TransE, researchers have proposed several improved models, including TransH (Wang et al., 2014), TransR (Lin et al., 2015), TransD (Ji et al., 2015), TransG (Huang et al., 2022), and RotatE (Sun et al., 2019). The first four models respectively incorporate hyperplanes, relationship spaces, dynamic transformation matrices, and relationship clustering to address the shortcomings of the translation model. RotatE defines relationships as rotations from the head entity to the tail entity in a complex vector space, enabling more effective inference of relationship patterns like symmetry, antisymmetry, and inversion.

Semantic matching-based models evaluate the confidence of factual triples by setting a scoring function to measure the semantic similarity between entities or relationships. RESCAL (Nickel, Tresp & Kriegel, 2011), the earliest semantic matching model, models complex relationships using three-dimensional tensor decomposition. Each relationship is represented as a slice in the tensor, capturing the interactions between the latent factors of the head and tail entities. However, this approach results in high model complexity due to the large number of parameters. DistMult (Yang et al., 2014) simplifies RESCAL by using a diagonal matrix to represent relationships, reducing the number of parameters and improving training efficiency. However, due to the limitations of diagonal matrices, DistMult is more suited for modeling symmetric relationships and has limited capacity for modeling asymmetric ones. Holographic embedding (HolE) (Nickel, Rosasco & Poggio, 2016) combines the advantages of RESCAL and DistMult, achieving a balance between expressiveness and efficiency by introducing cyclic correlation operations to capture complex relationship interactions between entities. Complex Embeddings (ComplEx) (Trouillon et al., 2016) represents entities and relationships as complex vectors, using complex multiplication and conjugation to calculate the compatibility between head entities, relationships, and tail entities. The plausibility of triples is assessed by defining a scoring function in the complex space.

Graph convolutional networks (GCNs) are deep learning models designed for processing non-Euclidean graph data. They extract useful feature representations from the graph structure through neighborhood aggregation and feature updates. ConvE (Dettmers et al., 2018) introduces two-dimensional convolution into KG embedding, exploring additional feature interactions through the joint representation of head entities and relationships, but does not explicitly incorporate tail entity information. ConvKB (Nguyen et al., 2017) expands the field of vision in design by jointly embedding head entities, relationships, and tail entities, capturing more semantic associations through convolution operations. Vectorized relational graph convolutional network (VR-GCN) (Ye et al., 2019) only updates entity embeddings without considering relationship embeddings. CompGCN (Vashishth et al., 2020) uses composition operators, TransGCN (Cai et al., 2019) introduces a new heterogeneous neighborhood representation method. Although they both learn embeddings for entities and relationships simultaneously, they do not take into account the feature representation of neighboring entities when updating relationships. Multi-channel convolutional model for knowledge graph completion (MConvKGC) (Sun et al., 2024) employs three separate feature extraction channels to concurrently capture shallow semantics, latent interactions, and translational characteristics. It utilizes attention mechanisms on these feature maps to further learn the local dependencies between entities and relationships. CTKGC (Feng et al., 2022) performs vector multiplication on subject embedding and relationship embedding to generate a 2D matrix and achieve full fusion of embedding at the element level, but it has limited ability to handle sparse data.

Attention mechanism-based KGC models have been proposed. MRGAT (Dai et al., 2022) can adapt to different cases of heterogeneous multi-relational connections and learns the representations of entities and relationships by a graph attention networks based framework. Learnable convolutional attention network for knowledge graph completion (LCA-KGC) (Shang, Zhao & Liu, 2024) introduces a learnable KG convolutional attention network that learns the amount of attention required for each local structure. Hyperbolic Hierarchical Attention Network model for Knowledge Graph Completion (HHAN-KGC) (Luo & Song, 2024) leverages the properties of hyperbolic space to embed complex relational patterns in lower dimensions, effectively handling various types of relationships in the KG. GATH (Wei, Song & Yao, 2024) is a graph attention network designed for heterogeneous KGC. It incorporates an entity-specific attention network and an entity-relationship joint attention network to address overfitting in sparse relationships and improve the distinction of entities sharing the same relation. By dynamically adjusting attention weights and jointly modeling entity-relationship interactions, GATH effectively captures complex relational patterns. However, attention mechanism-based models often simplify relationships when aggregating neighbor information in KGs, potentially overlooking their complex semantics and features (Dai et al., 2022; Shang, Zhao & Liu, 2024).

Multi-relational graph neural architecture search (MR-GNAS) (Zheng et al., 2022) introduces an architecture search framework for multi-relational graph neural networks (GNNs), automating the design of highly effective architectures. It enhances flexibility through a fine-grained message passing schema, supporting diverse entity and relation types. Chen et al. (2021) proposed TACT (Topology-aware correlations), which categorizes all relation pairs into six topological patterns and converts an original KG into a RCG. On the RCG, all correlation coefficients between relationships are learned and aggregated with attention network to get neighborhood embeddings. Then a relationship’s embedding is obtained by splicing the neighborhood embedding with the initial embedding. Finally TACT uses a graph structural network to embed local graphs into vectors based on GraIL (Teru, Denis & Hamilton, 2020). ComDensE (Minsang Kim, 2022) combines relationship-aware and common features using dense neural networks. In the relationship-aware feature extraction, it attempts to create relational inductive bias by applying an encoding function specific to each relationship. However, in these models, the embeddings of entities and relationships are updated separately, limiting their ability to leverage each other’s information during updates.

Proposed method

Currently, many KGC methods based on GCNs mainly focus on updating entities representations by aggregating the neighbor information (Ye et al., 2019; Shang, Zhao & Liu, 2024). Although some methods update relationship representations, they do not apply local neighborhood entity information (Vashishth et al., 2020; Cai et al., 2019). These methods often limit their performance in handling higher-order relationships and more complex patterns in multi-relational graphs.

To address the challenges and limitations of the aforementioned methods, this article proposes JGURD, which jointly updates entities and relationships based on the directional information of relationships in KG. Figure 2 illustrates the overall architecture of JGURD. Part 1 converts a KG into graph structure and randomly initializes the embeddings of entities and relationships using a Gaussian distribution. Meanwhile, the KG is transformed into a RCG, and relationship direction embeddings are generated through fully connected neural network layers (FCNN). The relationship direction embeddings are then be spliced to entities’ randomly initialized embedding as the entities’ relation-aware embeddings. Part 2 encodes entities and relationships. Entities are encoded by concatenating Gaussian-initialized embeddings with relationship direction embeddings to form relation-aware embeddings, which are fed into a GCN operating on the KG. This allows entities to capture structural information from their neighbors. Relationships are encoded using a GCN combined with a multi-layer attention mechanism on the RCG. The attention mechanism includes relationship similarity computation, attention weight assignment, and weighted summation, enabling effective modeling of relational interactions. Part 3 performs joint gradient updates for entities and relationships. A scoring function evaluates the quality of triples, and gradients are propagated to simultaneously update embeddings. Entity embeddings are updated based on connected relationships and neighboring entities, while relationship embeddings incorporate information from connected entities and other relationships. This joint update mechanism ensures mutual interaction between embeddings, capturing the complex dynamics of the KG. Part 4 processes the updated embeddings using three decoders—TransE, RotatE, and DistMult—to evaluate the embeddings and perform downstream KGC task.

Figure 2 An overview of JGURD.

Construction of KG and RCG

A KG is represented as a directed multi-relational graph, where nodes represent entities, edges represent relationships between entities. The graph is a multi-typed edge structure that supports multiple edge types between the same node pairs and allows self-loops (i.e., nodes can connect to themselves through specific relationships). A triple is converted into a graph by mapping the head and tail entities to two nodes, with the relationship represented as a directed edge from the head to the tail. For multiple relationships or self-loops, corresponding edges are added to ensure the graph fully captures the semantic and topological structure of the KG.

To model topological patterns between any two adjacent relationships, this article categorizes all adjacent relationships into six topological patterns, as illustrated in Fig. 3. The left part of the figure shows two relationships from adjacent triples and their connection to central entity, with arrows indicating the directionality of relationships. The topological pattern between r1 and r2 is determined by their directional connection to the central entity. For instance, if r1 is an incoming relationship to the central entity and r2 is an outcoming relationship from the central entity, the pattern is Input-to-Output (I-O). Six relational topological patterns are identified: Input-to-Output (I-O), Output-to-Output (O-O), Input-to-Input (I-I), Output-to-Input (O-I), Similar (SIM), and Inverse (INV). All adjacent triples fit into one of these categories. The right part of the figure illustrates their corresponding representations in the RCG obtained from the transformed KG, where nodes r1 and r2 represent the relationships from the KG, and the labeled arrow from r1 to r2 denotes topological pattern.

Figure 3 Six topological patterns.

To facilitate the learning of relationship embeddings, an original KG is transformed into RCG based on the classification of topological patterns between relationships. As shown Fig. 4, the left side of the figure illustrates a subgraph of the original KG, where u, v, e1, and e2 represent entities, and r1,r2,r3,r4,r5,r6 represent relationships. The dashed line indicates the central node rt, which represents a specific relationship in the original KG. The right side shows the transformed RCG, where nodes represent relationships from the original KG, rather than entities. The central node rt connects to other nodes r1,r2,…,r6. The edges in the RCG capture the topological patterns between relationships, such as Input-to-Output (I-O), Output-to-Output (O-O), and so on, reflecting the directional connections between them.

Figure 4 Transformation from an original KG to a relationship correlation graph (RCG).

For a KG dataset, the construction process of the RCG is similar to that of the KG, except that nodes in RCG represent relationships from original KG rather than entities. In the construction of RCG, all relationships are first extracted from the KG and added as nodes in the RCG. Then, by analyzing the topological patterns between relationships, edges are established between nodes to reflect the directional connections between relationships. During this process, relationship patterns that may cause self-loops are excluded, ensuring that no self-loops appear in the RCG.

Multi-relation and entity encoding mechanism

After KG and RCG are represented as graph structures, the vector representations of each entity and relationship are initialized as the starting point for model learning. This article employs a Gaussian distribution (Wu et al., 2024; Fei et al., 2024) to randomly initialize embeddings for entities and relationships, ensuring randomness and diversity. Although the initial embeddings do not capture semantic and structural information of KGs, they provide the basic input for subsequent learning. The randomly initialized embeddings are refined by encoders and iteratively optimized during model training to more accurately capture the semantic relationships and topological features of KG.

Multi-relation encoder

Based on the idea of TACT, we encode relationships on the RCG, where the nodes on the RCG represent relationships in the original KG and the edges represent the topological patterns between two relationships. Unlike TACT, which employs an attention network to generate relationship embeddings, we propose a novel relationship encoder that integrates graph convolutional networks (GCN) with a multi-layer attention mechanism. In our method, GCN aggregates local information through convolution operations, focusing on relationships with 1-hop neighbors to extract local features. Meanwhile, multi-layer attention mechanism progressively expands the receptive field, capturing relationships at greater distances (e.g., 2-hop, 3-hop) to extract global features. This allows our encoder to capture both local and global features, effectively modeling the complex dependencies between relationships. Additionally, a dropout layer is added after the output of each layer in the multi-layer attention to alleviate overfitting issue.

Firstly, GCN layers are employed to derive the local structure encoding rv for relationships, as shown in Eq. (1).

(1) rv(l)=σ(∑u∈N(v)1cvuW(l)r~u(l−1)+b(l))

where rv(l) denotes the feature representation of node v at the l-th layer, cvu=|N(v)|×|N(u)| is symmetric normalization coefficient between nodes v and u, N(v) and N(u) is the set of neighbor nodes of node v and u. W(l) and b(l) represent the weight matrix and bias vector, respectively, both of which are learnable parameters in neural networks. Similarly, W and b mentioned below are also learnable parameters, but their specific values depend on the context. σ is ReLU activation function.

We introduce a multi-layer attention mechanism to capture the attention scores between relationships (nodes in the RCG) connected through six topological patterns and obtain the global structure encoding of relationships r~v. The calculation process is shown in Eqs. (2)–(5).

(2) euv(l)=LeakyReLU(a(l)T[W(l)ru(l−1)||W(l)rv(l−1)])

(3) e~uv(l)=Dropout(euv(l),DropoutRate)

(4) αuv(l)=exp⁡(e~uv(l))∑w∈N(v)exp(e~uv(l))

(5) r~v(l)=σ(∑u∈N(v)αuv(l)W(l)ru(l−1))

where euv(l) is attention score between nodes v and u, DropoutRate is dropout ratio, αuv(l) is softmax-normalized attention weight, LeakyReLU is activation function, a(l) presents parameter vector of the attention mechanism, and || denotes concatenation operation.

Furthermore, local and global structure encodings are fused through feature concatenation and processed with linear transformation and nonlinear activation to obtain the final relationship representation er.

(6) er=σ([rv⊕r~v]W)

where ⊕ denotes matrix concatenation operation. Note that during relationship encoding phase, the nodes represent relationships in KGs, which correspond to the nodes in RCGs.

Relation direction-aware entity encoder

Relation direction-aware entity encoding aims to enhance entity representations by leveraging the directional features of relationships and local structure information from the KG, enabling more precise capture of entity semantics. Therefore, we design a FCNN that consists of a hidden layer and an output layer, with direction labels as input and relationship direction embeddings as output, as depicted in Eqs. (7) and (8). Note that the process of obtaining relationship direction embeddings and entity encoding is conducted on original KG.

(7) K(l)=ReLU(XWh+bh)

(8) er(dir)=K(l)Wo+bo

where X is input direction label (“incoming” or “outgoing”), represented as a vector generated via one-hot encoding. er(dir) is the final relationship direction embedding. dir∈{in,out}, where in indicates the direction in which a relationship points to an entity (incoming), and out indicates the direction in which a relationship emanates from an entity (outgoing). K(l) is hidden feature representation of the l-th layer in the network, where l denotes layer index. The weight matrix Wh maps input directional label X to the hidden feature space. As a learnable parameter, Wh captures linear transformation between directional labels and hidden features. Similarly, bh, the bias vector, is another learnable parameter that adjusts the output of linear transformation and introduces nonlinearity, thereby enhancing the model’s expressive power. The roles of Wo and bo are analogous.

The relationship direction embeddings of a given entity are pooled separately and then concatenated. || denotes the concatenation operation.

(9) eravg=1|Nin(v)|∑r∈Nin(v)er(in)||1|Nout(v)|∑r∈Nout(v)er(out).

The pooled result eravg is concatenated with the entity’s randomly initialized embedding vector eu derived from a Gaussian distribution as relation-aware entity embedding e¯u, shown in Eq. (10). e¯u integrates multi-relational directional information and serves as the input to the encoder. || denotes the concatenation operation.

(10) e¯u=eu∥eravg.

GCN captures entity features primarily from the direct neighborhood information of entities, disregarding the global information of the entire graph. Similar to relationship encoding, convolutional layer dropout is incorporated. The relation-aware entity embedding e¯u serves as the initial input to the encoder, with ei(0)=e¯u for each entity i at the zeroth layer. In subsequent layers ( l>0), the embeddings are refined through neighborhood aggregation, as defined in Eqs. (11) and (12). Specifically, at each layer l, the encoding ei(l) for node i is updated based on Eqs. (11) and (12).

(11) e~i(l)=σ(∑j∈N(i)1deg(i)⋅deg(j)ej(l−1)W(l))

(12) ei(l)=Dropout(e~i(l),DropoutRate)

where ei represents the final embedding of an encoded entity. N(i) represents the neighbors of node i, deg(i) denotes the degree of node i, and σ(⋅) represents the ReLU activation function. Note that the nodes in the entity encoding process represent the entities in KGs.

Gradient descent update rule

Figure 5 illustrates that VR-GCN, which uses a vector relationship GCN, only updates entity embeddings without considering relationship embeddings. CompGCN uses composition operators, TransGCN introduces a new heterogeneous neighborhood representation method. Although they both learn embeddings for entities and relationships simultaneously, they do not take into account the feature representation of neighbor entities when updating relationships. The proposed JGURD employs convolutional operations for both entity and relationship feature representations. Additionally, it incorporates entity neighborhood information for relationship feature updates and defines similar updating methods for both entities and relationships. Note that during the joint gradient update process, nodes denote entities in KGs.

Figure 5 The distinctions between JGURD and relevant models, where eul and erl denote the un-updated representations of entities and relationships, respectively.

* denotes the graph convolution operation, W denotes the model parameters of the linear transformation, and σ denotes the activation function.

Updating entity and relationship embeddings jointly with similar methods improve model consistency and interpretability. JGURD introduces scoring function f(⋅) to evaluate triple rationality, which is crucial for identifying factual triples. The embedding and updating of multi-layer nodes are shown in Eqs. (13) and (14).

(13) mv(l+1)=∑u∈N(v)∂f(eu(l),ev(l))∂ev(l)

(14) ev(l+1)=σ(W(l)(mv(l+1)+ev(l)))

where ev(l) denotes the embedding of node v at layer l, N(v) denotes the set of neighbor nodes of v, mv(l+1) represents the aggregated representation of these neighbors. Constraining f(⋅) to be the inner product of the head and tail entities, i.e., f(eu,ev)=euTev, the gradient of mv(l+1)+ev(l) will increase further, and the sum of the triple scoring functions within the neighborhood, ∑f(eu,ev), can be maximized.

In multi-relational graphs, relationship embedding is crucial for the scoring function that measures the rationality of triples, which is often ignored by most GCN-based KGC methods. To apply similar update rules and methods for entities and relationships, their feature representations should be learned jointly. By incorporating relationship embeddings and categorizing the directions of relationship as incoming and outgoing, the entity update rule in Eq. (13) is defined as follows:

(15) mv(l+1)=∑(u,r)∈Nin(v)We(l)∂fin(eu(l),er(l),ev(l))∂ev(l)+∑(u,r)∈Nout(v)We(l)∂fout(ev(l),er(l),eu(l))∂ev(l)

(16) ev(l+1)=σ(mv(l+1)+Wse(l)ev(l)).

In Eq. (15), Nin(v) denotes the set of incoming neighbors for node v, and Nout(v) denotes the set of outgoing neighbors for node v. Wse is the self-loop update weight matrix for an entity. Equation (16) indicates that the entity’s final feature representation is derived from the message passing and self-loop message.

As illustrated in Fig. 6, the blue entities represent the entities within the Nin(v) set, while the purple entities represent the entities within the Nout(v) set. Similarly, the relationship update rule, which takes into account entity neighborhood information, is defined as follows:

(17) mr(l+1)=∑(u,v)∈N(r)Wr(l)∂fr(eu(l),er(l),ev(l))∂er(l)

(18) er(l+1)=σ(mr(l+1)+Wsr(l)er(l))

where N(r) denotes the direct neighbor set of relationship r, including the head entity and tail entity in the triple. Wsr is the self-loop update weight matrix of a relationship.

Figure 6 The set of incoming and outgoing neighbors of the central entity.

Figure 7 illustrates the updating process of JGURD. On the left side of the figure, the scoring function f(⋅) is used to evaluate the quality of each triple, where each triple consists of entity nodes (e.g., u1,u2,u3) and a relationship (e.g., r1,r2,r3). This scoring function calculates the score for each triple and serves as the basis for updating the embeddings of both entities and relationships. The right side of the figure shows how the gradients of the scoring function are computed and used to update the embeddings. Specifically: Fv=∂f/∂v is the gradient used to update the embedding of the central entity v. It is calculated based on the influence of its connected relationships r1,r2,r3 and neighboring entity (e.g., u1,u2,u3).

For the relationship rk, its gradient Frk=∂f/∂rk depends not only on the relationship itself but also on the embeddings of the entity connected to it. Specifically, the update of rk is influenced by the gradient propagation from its connected entity nodes (e.g., uk and v), where k=1,2,3,…. This ensures that the update of the relationship reflects the influence of entities.

For the entity uk, its gradient Fuk=∂f/∂uk represents the interactions between the entity node, its associated relationship rk, and other entity nodes, where k=1,2,3,….

Figure 7 Joint gradient update process.

This joint gradient update mechanism ensures mutual interaction between the embeddings of entities and relationships during the update process.

To achieve training consistency, this article makes fin=fout=fr in the calculation of triple score and uses a uniform weight matrix, We(l)=Wr(l). In graph neural networks, entity and relationship node representations tend to be dominated by neighbor node information through iterative message passing and updates, potentially leading to distortion in node representation. To tackle this issue, different self-loop update weight matrices are added for the entities and relationships, that is, Wse(l)≠Wsr(l). This ensures their self-information can be preserved in each update round, enhancing the robustness of their representations. To prevent numerical instability, gradient explosion, and gradient vanishing, we introduced normalization factors for both entities and relationships. Replace mv(l+1) with αmv(l+1)/(|Nin(v)|+|Nout(v)|) in Eq. (16). Replace mr(l+1) with βmr(l+1)/|N(r)| in Eq. (18). |⋅| represents the number of elements in set, α and β are hyperparameters controlling the weight of aggregated neighboring information.

Multi-decoder layer

Different decoders employ diverse scoring and loss functions, enriching a model’s expressive capabilities and enhancing its understanding of complex relationships between entities. This article attempts three decoders, including TransE, RotatE, and DistMult. They are individually designed, implemented, refined, and integrated into the entire architecture of JGURD to find the most suitable one. Decoder 1: JGURD-TransE

As a kind of distance translation-based model, TransE assums that the relationship serves as the translation between the head entity and the tail entity. For the triple (eu,er,ev), scoring function is defined by Eq. (19).

(19) fer(eu,ev)=−∥eu+er−ev∥

where eu, er, and ev represent the head entity vector, relationship vector, and tail entity vector, respectively. To further optimize JGURD, this article adopts the Bernoulli negative sampling (Yang et al., 2024; Yao et al., 2023) method to construct a certain number of negative samples. The margin-based ranking loss function is shown in Eq. (20).

(20) L=∑(eu,er,ev)∈T∑(eu′,er,ev′)∈T′max(0,−fer(eu,ev)+fer(eu′,ev′)+γ)

where T denotes the set of factual triples, T′ represents the set of negative triples, max(a,b) represents the maximum value of a and b, and γ is the margin parameter. Decoder 2: JGURD-RotatE

The second decoder is based on rotational hypothesis. RotatE incorporates knowledge from complex vector spaces, rotating entities within this space, and deriving tail entities from head entities. The scoring function is defined by Eq. (21).

(21) fer(eu,ev)=−∥eu⊙er−ev∥

Self-adversarial negative sampling is used to train JGURD, it employs a probability distribution to generate negative samples (Kamigaito & Hayashi, 2022). Assuming (ui′,er,vi′) is the negative sample for the triple (ui,er,vi), the probability distribution p for other negative sample triples (ui′,er,vi′) is given by Eq. (22).

(22) p(uj′,er,vj′)=exp⁡(αfer(uj′,vj′))∑(ui′,er,vi′)∈T′exp(αfer(ui′,vi′))

where α is a constant. Based on the probability weights of the aforementioned negative samples, the loss function is set as shown in Eq. (23). σ denotes the sigmoid activation function, and the embeddings of entities and relationships are both in the complex vector space.

(23) L=−log⁡(σ(γ+fer(eu,ev)))−∑(eu′,er,ev′)∈T′p(eu′,er,ev′)log⁡(σ(−fer(eu′,ev′)−γ)).

Decoder 3: JGURD-DistMult

Lastly, based on the hypothesis of bilinear scoring function, DistMult enhances the connections between entities and relationships. It utilizes a two-dimensional diagonal matrix operator Mer to replace the tensor matrix. The scoring function is shown in Eq. (24).

(24) fer(eu,ev)=euTMerev.

The Bernoulli negative sampling method is used to construct negative samples, and the minimized cross-entropy loss function is shown in Eq. (25).

(25) L=−1(1+ω)|ε′|∑(eu,er,ev)∈Tylog⁡σ(fer(eu,ev))+(1−y)log⁡(1−σ(fer(eu,ev)))

where ω is the number of negative samples, ε′ is the entity sampling range, and y is the label added to factual triples and negative sample triples (1 or 0). σ(⋅) denotes the sigmoid activation function.

Training

This article employs the point-wise cross-entropy loss function (Ji et al., 2021) to update stage parameters. It disregards sample order and introduces probability p from the binomial distribution to label positive and negative triples. By minimizing the loss function, the scores of positive triples are maximized while the scores of negative triples are minimized. The loss function is shown in Eq. (26).

(26) L=−1S∑(eu,er,ev)∈T∑i=1S[yilog⁡y^i+(1−yi)log⁡(1−y^i)]

where S represents the batch size of training, T is the set of triples, yi denotes the label of triple fi(eu,er,ev), taking values of 1 or 0, and y^i denotes the mapping result after applying the softmax function.

(27) y^i=g(fi(eu,er,ev))

where g(⋅) denotes the softmax function and f(⋅) denotes the score function of the triple.

Results

Datasets and evaluation metrics

This article carries out experiments on FB15K (Bordes et al., 2013), FB15k-237 (Bordes et al., 2013), WN18RR (Dettmers et al., 2018), and NELL-995 (Xiong et al., 2018) datasets. Models can predict most triples effortlessly when there are numerous reversible relationships in a dataset, bring excessive evaluation benchmarks. To avoid this issue, the FB15k-237 and WN18RR datasets were created by eliminating the inverse relationships in FB15k and WN18. WN18 dataset URL is https://paperswithcode.com/dataset/wn18 and FB15k dataset URL is https://everest.hds.utc.fr/doku.php?id=en:transe. The NELL-995 dataset is derived from the 995th iteration of the NELL system, consisting of triples with the 200 most frequent relationships. NELL-995 dataset URL is https://rtw.ml.cmu.edu/rtw/. As shown in Table 1, this article divides datasets into training, validation, and test sets. |V| and |E| represent the number of entities and relationships, respectively.

Table 1 Description of the datasets.

Dataset	|V|	|E|	Training	Validation	Test	
FB15k-237	14,541	237	272,115	17,535	20,466	
FB15k	14,951	1,345	483,142	50,000	59,071	
WN18RR	40,943	11	86,835	3,034	3,134	
NELL-995	75,491	200	137,645	2,628	2,601	

We utilize the PyTorch and DGL frameworks, employing the Adam optimizer for gradient descent algorithm. The vector dimensions are set to 100 for datasets FB15k, FB15k-237, and NELL-995, and 50 for WN18RR. The learning rate is 0.001, batch size is 64, encoder training epochs are 500, dropout rate is 0.3. The number of graph convolutional layers is set to 1, while the number of attention layers is set to 3.

In our experimental evaluation, we employed mean reciprocal rank (MRR) and the Hits@N metric, with N values of 1, 3, and 10.

Figure 8 is a complete workflow diagram to illustrate how data is processed, split, and used during training, validation, and testing. The original dataset was cleaned and checked for outliers, and subsequently split into training, validation, and test sets. As the dataset is publicly available, the splitting has been previously performed by other researchers following a common proportion (Wei, Song & Yao, 2024; Luo & Song, 2024). The link to the partitioned data is https://github.com/Lemen569/rawdata_JGURD. The details of the split are provided in Table 1. The training set, after applying different negative sampling strategies, uses a subset of negative samples along with positive samples for model training. The validation set is then used for hyperparameter tuning and model validation, while the test set is used for final model testing. The model performance is ultimately evaluated through the Hits@1, 3, 10, and MRR metrics.

Figure 8 A complete workflow diagram from raw dataset processing to model performance evaluation.

In training, negative sample generation is crucial for model training. It occurs before each training epoch, during the data preparation phase, ensuring that both negative and positive samples are included in the training process, rather than being dynamically generated during model training or evaluation. From the perspective of the generation method, the Bernoulli negative sampling strategy is used for TransE and DistMult decoders. Based on the relation characteristics of each positive sample (e.g., 1-N or N-1), the Bernoulli distribution determines whether to replace a head or tail entity. Each positive sample generates multiple negative samples to enhance training diversity. For the RotatE decoder, a model-based self-adversarial negative sampling strategy is employed. Candidate negative samples are scored, and those with higher scores are selected based on the softmax-derived distribution. In addition, the generation of negative samples is restricted to the training set. During the entire training process, the validation and test sets are not involved in negative sample generation, ensuring their independence and effectively preventing data leakage.

Comparison experiments

We compare JGURD with 14 related models by experiments on four public datasets and the results are listed in Tables 2 and 3. The bold number presents the best result, the underlined number presents the second-ranked result, and the dash indicates the original reference didn’t supply according experiment result.

Table 2 Comparison of each indicator for the experiments conducted on WN18RR and FB15k.

Numbers in bold present the best result, the underlined numbers present the second-ranked result.

Model	WN18RR	FB15k	
	Hits@1	Hits@3	Hits@10	MRR	Hits@1	Hits@3	Hits@10	MRR	
TransE	0.415	0.484	0.465	0.494	0.197	0.278	0.249	0.163	
DistMult	0.555	0.463	0.519	0.541	0.103	0.118	0.114	0.334	
ConvE	0.437	0.356	0.401	0.525	0.258	0.323	0.331	0.357	
LCA-KGC	0.456	0.510	0.585	0.492	–	–	–	–	
SACN	0.560	0.590	0.542	0.572	0.408	0.449	0.447	0.402	
TACT	0.982	0.942	0.956	0.988	0.410	0.395	0.484	0.542	
ConvKB	0.455	0.571	0.521	0.443	0.373	0.295	0.225	0.497	
MConvKGC	0.428	0.479	0.545	0.466	–	–	–	–	
MRGAT	0.443	0.501	0.568	0.481	–	–	–	–	
MR-GNAS	0.410	0.470	0.541	0.456	–	–	–	–	
GATH	0.426	0.475	0.537	0.463	–	–	–	–	
CTKGC	0.426	0.472	0.521	0.459	–	–	–	–	
ComDensE	0.440	–	0.538	0.473	–	–	–	–	
HHAN-KGC	0.522	0.503	0.630	0.587	0.486	0.533	0.589	0.504	
JGURD-DistMult	0.497	0.573	0.544	0.565	0.536	0.429	0.544	0.526	
JGURD-RotatE	0.570	0.596	0.555	0.566	0.573	0.569	0.584	0.549	
JGURD-TransE	0.593	0.512	0.540	0.535	0.542	0.558	0.472	0.537	

Table 3 Comparison of each indicator for the experiments conducted on FB15k-237 and NELL-995.

Numbers in bold present the best result, the underlined numbers present the second-ranked result.

Model	FB15k-237	NELL-995	
	Hits@1	Hits@3	Hits@10	MRR	Hits@1	Hits@3	Hits@10	MRR	
TransE	0.197	0.478	0.349	0.363	0.118	0.168	0.253	0.191	
DistMult	0.222	0.218	0.314	0.234	0.186	0.198	0.164	0.238	
ConvE	0.258	0.223	0.331	0.357	0.301	0.137	0.253	0.210	
LCA-KGC	0.276	0.407	0.554	0.372	–	–	–	–	
SACN	0.438	0.449	0.447	0.402	0.503	0.465	0.560	0.461	
TACT	0.378	0.452	0.549	0.575	0.444	0.413	0.540	0.571	
ConvKB	0.273	0.295	0.225	0.297	0.360	0.460	0.336	0.359	
MConvKGC	0.257	0.382	0.535	0.348	–	–	–	–	
MRGAT	0.266	0.386	0.542	0.358	–	–	–	–	
MR-GNAS	0.258	0.380	0.530	0.348	–	–	–	–	
GATH	0.253	0.376	0.527	0.344	–	–	–	–	
CTKGC	0.252	0.372	0.517	0.340	0.349	0.433	0.496	0.403	
ComDensE	0.265	–	0.536	0.356	0.348	–	0.536	0.356	
HHAN-KGC	0.260	0.425	0.528	0.408	0.495	0.586	0.698	0.582	
JGURD-DistMult	0.546	0.501	0.552	0.572	0.648	0.629	0.426	0.495	
JGURD-RotatE	0.570	0.533	0.585	0.592	0.653	0.698	0.668	0.596	
JGURD-TransE	0.553	0.519	0.529	0.540	0.650	0.610	0.542	0.475	

We can observe that JGURD overall outperforms other models across all metrics. On the FB15k dataset, the JGURD achieves 6.8% and 8.9%, higher results than the second-place method (HHAN-KGC) on Hits@3 and MRR, respectively. This advantage may stem from JGURD’s multi-relation encoding mechanism, which models complex relational associations via RCG and enhances entity feature learning through a relation-aware entity encoder. In contrast to HHAN-KGC, which uses hyperbolic space and hierarchical attention mechanisms to model complex relations, JGURD shows clear superiority in handling intricate relational patterns. Meanwhile, the FB15k dataset contains 1,345 relations and diverse relational patterns (1-N, N-1, N-N), providing ample modeling space for JGURD’s RCG mechanism and further enhancing its performance.

In addition, both JGURD and TACT use RCG, the original reference of TACT only supplied the values of Hits@1 and MRR on two datasets, i.e., WN18RR and FB15k-237. To compare JGURD and TACT comprehensively, we have complemented TACT’s Hits@3 and Hist@10 on WN18RR and FB15k-237, and Hit@1, Hit@3, Hit@10 and MRR on FB15k and NELL-995 in Tables 2 and 3. JGURD is superior to TACT on all three datasets except WN18RR. It is because the scarcity of relationships in WN18RR cannot supply sufficient information in the topological structure of relationships when JGURD aggregates relationships in the RCG. However, on other three datasets, JGURD demonstrates a significant advantage. For instance, compared to TACT, JGURD showed increases of 17.9%, 6.6%, and 3.0% in Hits@3, Hits@10, and MRR on the FB15k-237 dataset, respectively, indicating that the performance improvement of JGURD is primarily due to its model design. For example, JGURD’s joint gradient update mechanism enhances the model’s ability to capture dynamic entity-relationship interactions and enables efficient adaptation to the heterogeneous structures and feature distributions of different datasets. This adaptability allows JGURD to achieve significant performance gains across various datasets. On datasets like FB15k-237 and NELL-995, JGURD outperforms all other methods. Even on the relation-sparse WN18RR dataset, its innovative framework outperforms most comparison models. In the WN18RR dataset, its joint gradient update mechanism extracts more meaningful feature representations from sparse relational data, enhancing its robustness and accuracy in completion tasks.

Moreover, some baseline models exhibit significant performance differences across datasets with sparse relationships (WN18RR) and those with denser relationships (FB15k, FB15k-237, and NELL-995). For instance, models like SACN, TACT and CTKGC perform notably worse on the other three datasets compared to WN18RR, indicating they are heavily influenced by dataset distribution. In contrast, our JGURD model shows smaller performance differences across these four datasets and is less affected by distribution characteristics, exhibiting relatively better generalization ability.

Finally, as shown in Tables 2 and 3, the RotatE decoder in JGURD’s multi-decoder mechanism outperforms the others. This is mainly due to its ability to model complex relational patterns effectively through rotation operations in the complex plane, particularly excelling on datasets such as FB15k-237 and FB15k, which include symmetric and antisymmetric relationships. Additionally, the TransE and DistMult decoders each have distinct advantages: TransE excels at modeling one-to-one relations, while DistMult performs well with symmetric relations. This multi-scoring function strategy allows JGURD to adapt flexibly to the feature distributions of different datasets, thereby enhancing its robustness.

To further validate the effectiveness of the joint gradient update method put forward by us, we also compare JGURD with VR-GCN, CompGCN, and TransGCN on FB15k-237. As shown in Table 4, JGURD perform best. In particular, JGURD-RotatE achieves a Hits@10 improvement of approximately 9.3% over CompGCN and 5.4% over TransGCN. This advantage stems from JGURD’s joint gradient update mechanism, which integrates local neighborhood entity information during relationship updates. Through iterative updates using forward propagation gradients of entities and relationships, it facilitates dynamic interaction and collaboration between entity and relationship representations, optimizing model parameters. In contrast, while CompGCN and TransGCN jointly consider entities and relationships using combination operators, they do not leverage entity feature representations in the neighborhood to update relationships. This weakens the coordination between entity and relationship representations, limiting model performance.

Table 4 Validation of the joint gradient update method.

Numbers in bold present the best result, the underlined numbers present the second-ranked result.

Model	FB15k-237	
	Hits@1	Hits@3	Hits@10	MRR	
VR-GCN	0.159	0.272	0.432	0.248	
CompGCN	0.264	0.390	0.535	0.355	
TransGCN	0.252	0.388	0.555	0.356	
JGURD-DistMult	0.546	0.501	0.552	0.572	
JGURD-RotatE	0.570	0.533	0.585	0.592	
JGURD-TransE	0.553	0.519	0.529	0.540	

Ablation experiments

The impact of GCN and multi-layer attention mechanism for multi-relation encoding

Relationship encoding employs a blend of GCN and multi-layer attention mechanism. This Ablation experiment is conducted on the FB15k-237 and NELL-995 datasets. “D,” “R,” and “T” represent the DistMult, RotatE, and TransE decoders, respectively. “1” signifies the exclusion of GCN, while “2” denotes the absence of multi-layer attention mechanism. For instance, “w/oJG-D1” denotes the JGURD-DistMult without GCN with all other settings identical to those of JGURD-DistMult, while “w/oJG-T2” indicates the JGURD-TransE without multi-layer attention mechanism with all other settings identical to those of JGURD-TransE.

As shown in Fig. 9, removing either GCN or multi-layer attention mechanism from the JGURD results in a certain degree of performance degradation. Compared to GCN, multi-layer attention mechanism has a more significant impact on performance. It’s because omitting the multi-layer attention mechanism would hinder JGURD’s ability to capture the global structure among relationship nodes in the RCG. Especially in multi-relational graph completion tasks, global information’s absence results in more significant performance degradation.

Figure 9 The Hits@1, Hits@3 results on FB15k-237 and NELL-995.

Impact of dropout rate

JGURD adds dropout for each layer in GCN and multi-layer attention mechanism. In this subsection, JGURD-RotatE with different dropout rates are conducted on four datasets to analyze the impact of dropout. Here, dropout rate is set as 0, 0.1, 0.3, 0.5, 0.7 and 0.8, respectively. Note that dropout is employed only in the training phase. It should be disabled in testing phase.

Figure 10 illustrates that performance of JGURD-RotatE improves steadily with dropout rates below 0.3, peaking at a dropout of 0.3. Subsequently, performance begins to decline as the dropout rate surpasses 0.3. This suggests that employing moderate dropout can attain a balanced regularization effect, effectively mitigating overfitting and enhancing generalization ability.

Figure 10 The Hits@1, Hits@3, Hits@10 results of different dropout rates.

Impact of relation directions

To explore the impact of the incoming and outgoing directions of relationships on KGC performance during entity encoding and joint gradient updates, this section presents two variants of JGURD and evaluates them on three multi-relation datasets: FB15k, FB15k-237, and NELL-995.

Variant A: In entity encoding and joint gradient updates the directions of relationships are ignored. Specifically entity encoding does not include relationship direction embeddings and only the randomly initialized entity embedding eu is used for graph convolution. During joint gradient updates the entity’s neighbor set is not differentiated by relationship direction.

Variant B: The incoming and outgoing directions of relationships are considered only during entity encoding, not during joint gradient updates.

The results are shown in Fig. 11, where D, R, and T denote JGURD-DistMult, JGURD-RotatE, and JGURD-TransE, respectively, and variant A and variant B are two variants of JGURD. It can be seen from the figure that considering the incoming and outgoing directions of relationships during entity encoding and joint gradient updates significantly affects KGC performance. Specifically, the MRR and Hits@10 of Variant A are lower than those of the original model across all datasets, indicating that completely ignoring relationship directions hinders the model’s ability to fully leverage the structural information in KGs, thereby impacting completion performance. Variant B performs better than Variant A but worse than the original model, indicating that considering relationship directions during entity encoding improves model performance to some extent. However, ignoring relationship directions during joint gradient updates still prevents the model from achieving optimal performance. This further emphasizes that relationship direction is a critical factor in KGC.

Figure 11 Comparison of JGURD variants with relationship direction consideration on the FB15k, NELL-995, and FB15k-237 datasets.

Conclusions

This article proposes JGURD to enhance the KGC for multi-relational graphs. JGURD uses GCNs and multi-layer attention mechanisms to encode relationships based on the topological structure of relation pairs, effectively capturing both local and global structural information. Incorporating the relational direction embedding vectors into the embedding of entities fully utilizes relational correlation information. Additionally, we put forward a joint gradient update method to simultaneously update entities and relationships, effectively alleviating the limitations of existing models when handling the interdependencies between entities and relationships. JGURD uses relational information for entity updates and, crucially, leverages local entity neighborhood information for relationship updates. Experimental results demonstrate that JGURD significantly outperforms other models in completing multi-relational KGs. JGURD primarily focuses on textual information and has not fully utilized external knowledge sources. In the future, we plan to explore effective ways to incorporate external knowledge sources for enhancing the method’s reasoning ability and to integrate multi-modal data into KGC.

Additional Information and Declarations

Competing Interests

The authors declare that they have no competing interests.

Author Contributions

Lianhong Ding conceived and designed the experiments, analyzed the data, authored or reviewed drafts of the article, and approved the final draft.

Mengxiao Li conceived and designed the experiments, performed the experiments, performed the computation work, prepared figures and/or tables, and approved the final draft.

Shengchang Gao performed the experiments, authored or reviewed drafts of the article, and approved the final draft.

Juntao Li analyzed the data, authored or reviewed drafts of the article, and approved the final draft.

Ruiping Yuan analyzed the data, authored or reviewed drafts of the article, and approved the final draft.

Jianye Yu conceived and designed the experiments, authored or reviewed drafts of the article, and approved the final draft.

Data Availability

The following information was supplied regarding data availability:

The raw data is available at GitHub and Zenodo

- https://github.com/Lemen569/rawdata_JGURD.git

- Lemen569. (2024). Lemen569/rawdata_JGURD: KGC (JGURD). Zenodo. https://doi.org/10.5281/zenodo.12541386

The third party data is available at:

- NELL-995 https://rtw.ml.cmu.edu/rtw/

- FB15k https://everest.hds.utc.fr/doku.php

- The FB15k-237 dataset is extracted from the FB15k dataset.

- WN18RR https://paperswithcode.com/dataset/wn18

The code is available at GitHub and Zenodo:

- https://github.com/Lemen569/Knowledge-Graph-Completion-Model-Based-on-Graph-Neural-Networks-1.0

- Lemen569. (2024). Lemen569/Knowledge-Graph-Completion-Model-Based-on-Graph-Neural-Networks-1.0: KGCJR (tag). Zenodo. https://doi.org/10.5281/zenodo.11637737.

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
