# Peer review of "JGURD: joint gradient update relational direction-enhanced method for knowledge graph completion"

_PeerJ Computer Science, doi:10.7717/peerj-cs.2808_

## Round 0.1 · original submission · Major Revisions

Dear Authors,

Your paper has been reviewed. Based on the reviewers' reports, significant revisions are needed before it is considered for publication in PEERJ Computer Science.

Please address the reviewers' concerns and resubmit your article once you have updated it. More precisely

1) The problem that the method proposed by the authors tries to solve must be clearly described.
2) All claims must be clearly demonstrated using the appropriate technical literature.
3) The description of the proposed method must be complete to avoid any inconsistencies.
4) The code and data should be available for the next round of review.

Reviewer 1 ·

Basic reporting

This is a review to paper “JGURD: Joint Gradient Update Relational Direction-Enhanced Method for Knowledge Graph Completion” submitted to PeerJ.

This is yet another paper that suggests some tweaks to improve performance on existing benchmarks for the knowledge graph completion problem. Over the last 5 years or so I have seen (and reviewed) dozens of such papers, which are more or less claim the same thing: a subset of existing systems perform slightly worse than our system on these benchmarks. Such state of the art suggests that to contribute to the field scientifically it is not enough to suggest yet another method that improves slightly the performance on existing benchmarks, it should be some more insightful and fundamental contribution. Unfortunately, this paper is written in such a manner that it is hard to judge whether this requirement is satisfied or not (however, I suspect that it is not). So, I recommend rejection, and give more specific reasons below.

1. The introduction makes a number of very strong and not properly supported (I would even say arrogant) claims:
- “KGC methods based on logical rules required computing a probabilistic network”: this is wrong, usual logical rules (e.g., Datalog rules) have nothing about probabilities and probabilistic networks and are widely used for KGC
- “making them [logical rules] unable to meet the demands of complex reasoning”: this is wrong, there are logic-based formalisms of very large (I would even say arbitrary, for practical purposes) variety of complexity, and there is no problem to find such a formalism that is able to perform complex reasoning, whatever is meant by ‘complex’ here
- “DistMult (Yang et al., 2014) is insensitive to complex relationships”: this makes no sense, because it is never said what is the difference between complex and not-so-complex (easy? simple?) relationships
- “DistMult (Yang et al., 2014) is insensitive to complex relationships, while ComplEx (Trouillon et al., 2016) and ConvE (Dettmers et al., 2018a) can model more flexibly, but increasing computational complexity”: I am quite confident that there is no paper that analyses computational complexity of these systems; moreover, I doubt that the latter two have higher computational complexity than the former: all of them are in polynomial time with low polynomial degree. It is of course a question whether any of them can be done better that polynomial (for example, in NC, or even in TC0 complexity), but again, I doubt that there is any difference here for these systems.
- “current research mainly focuses on simple undirected graphs”: this is wrong, all the hundreds (in not thousands) of papers on KGC focus on directed graphs with different types of edges, because KGs are essentially such graphs
- “This may result in GNNs failing to adequately capture global features”: this makes no sense, because it is never said what is a global feature
- “To address the poor performance of existing methods in completing knowledge graphs with complex relationships, this paper proposes JGURD”: as said before, it is not clear what is a complex relationship; moreover, no evidence is given for the claim that existing methods have poor performance on “in completing knowledge graphs with complex relationships” (whatever these graphs are)
- “relationships are encoded using both a GCN and a multi-layer attention mechanism on the RCG”: sorry, what is “the RCG”?
- (there many more)
- (In the similar category, but in Section PROPOSED METHOD) “Although some methods update relationship representations, they do not apply local neighborhood entity information (Vashishth et al., 2020; Cai et al., 2019). These methods often limit their performance in handling higher-order relationships and more complex patterns in multi-relational graphs.”: I do not quite understand what all this means exactly, but I do not see why, for example, INDIGO by Liu et al. (INDIGO: GNN-based inductive knowledge graph completion using pair-wise encoding, NeurIPS’21) does not satisfy these properties.

2. It is impossible to evaluate the contribution of the paper just because it is never properly formulated what is the problem its method tries to solve. The only place where the problem itself is introduced is the second paragraph of the introduction, which says “It [Knowledge Graph Completion] predicts the missing parts of a triple using known entities, relationships, attributes, or other information.” This is it, and the formal part of the paper goes straight to the method to solve this (pretty much unknown) problem.

3. The description of the proposed method is highly incomplete, inconsistent and does not make sense in many places. For example:
- “After representing the KG as a graph structure”: how exactly the KG is represented as a graph structure? Is the graph directed or undirected? Is it one type of edges or several? Are self-loops allowed? how triples are translated to parts of the graph?
- “Firstly, GCN layers are employed to derive the local structural encoding r_v for relationships, as shown in Eq. (1)”: First, Eq (1) does not mention r^v, it mentions only r_v^(l), and it is not clear at all how r_v and r_v^(l) are related. sSecond, v in both r_v^(l) and r_v is a node and l is a layer, while r is (as far as I understand) just says that r_v^(l) is a vector; so, how r_v can be an encoding for relationships, if it is indexed only by nodes?
- “c_vu represents the normalization coefficient between nodes v and u”: in English, article ‘the’ is used when what follows is already known for the reader (i.e., introduced before in the text, or there is one unique such thing in the whole world), but this is definitely not the case here, no normalization coefficients were discussed anywhere in the paper before. So, what is this coefficient? Same applies to the following “the weight matrix” and “the bias vector”.
- “Note that in the relationship encoding phase, the nodes involved denote the nodes (relationships in the KG) in the RCG”: I read this 10 times and thought about it for 10 minutes, but could not understand what this sentence is about.
- Equation (6) operates symbols, which are never mentioned anywhere else: \tilde r_v and W; to large extend, it is the same for4 r_v, which was mentioned, but it is unclear how it is actually computed.
- “Entity Encoder. To obtain initial entity embedding vectors …” Sorry, but these vectors are already initialised in line 147: “Gaussian distribution (Wu et al., 2024; Fei 148 et al., 2024) is employed in this paper to obtain randomly initialized embedding vectors for entities and relationships.” Why do we re-initialise them?
- Equations (7) and (8) again operate some symbols that are never introduced.

I pretty much stopped reading the paper at this point, because I am absolutely lost.

Experimental design

nothing new: evaluation of a new method on widely known benchmarks

Validity of the findings

could not understand the findings, see above

·

Basic reporting

1. It is necessary to place a period (.) at the end of the figure caption. Additionally, the description should be sufficient for the figure. For example, the descriptions of Figures 2 and 5 are not sufficient for the figures.

2. 1. References Section, “Bordes, A., Usunier, N., Garcia-Duran, A., Weston, J., and Yakhnenko, O. (2013a). Translating embeddings for modeling multi-relational data. Advances in neural information processing systems, 26. DOI 10.5555/2999792.2999923” is duplicated, and it’s better review again accurately all of references and citentions.

3. P.3/15, line 97: “Graph convolutional networks (GCNs)”, It's should be better to abbreviate words the first time we use them.

Experimental design

1. It would be better to make the code and data available for further review.

2. Why is the initial vector considered random? Since this network is a knowledge graph, each node and each edge has a specific meaning. Otherwise, it could be measured using models like Node2Vec, which take only the graph structure as input.

3. It is better to compare the proposed model with other link prediction-based models for multi-relational knowledge graphs (such as "DOI: 10.1109/ICDM54844.2022.00089", DOI: 10.1016/j.neunet.2022.07.014, etc.) so that the performance of the model can be better relied upon.

4. How does the RCG work, and how is it used in the proposed model? It would be better to explain it in more detail.

5. What is the meaning of 'six types of topological relationships'? Explain further and clarify.

Validity of the findings

no comment

---

## Round 0.2 · Major Revisions

Dear Authors,

Your paper has been reviewed. Based on the reviewers' reports, significant revisions are needed before it is considered for publication in PEERJ Computer Science.

Please address the reviewers' concerns and resubmit your article once you have updated it. More precisely:

1. A more detailed analysis of whether the performance variations are due to the structure and features of the dataset or the configuration of the models is necessary. Providing insights into how these factors contribute to model performance would strengthen the evaluation and make the conclusions more robust.

2. It is recommended to provide a clear workflow diagram and a detailed step-by-step explanation in a Python notebook (.ipynb format) to illustrate how data is processed, split, and used during training, validation, and testing.

3. The negative sampling strategy should be described more explicitly, including how and when negative samples are generated. Ensuring that negative samples are drawn only from the training set and not from the test set is essential to avoid data leakage.

·

Basic reporting

no comment.

Experimental design

no comment.

Validity of the findings

I appreciate the revisions made by the authors in response to previous comments. To further enhance the quality of the work and ensure greater transparency, I suggest considering the following points:

1. It is critical to ensure that the dataset distribution and other characteristics, as highlighted in Table1 and Table 3, are thoroughly analyzed to explain the performance differences between models. A more detailed analysis of whether the performance variations are due to the structure and features of the dataset or the configuration of the models is necessary. Providing insights into how these factors contribute to model performance would strengthen the evaluation and make the conclusions more robust.

2. There is a concern about potential data leakage based on the provided implementation, and this issue needs to be addressed with full transparency. It is recommended to provide a clear workflow diagram and a detailed step-by-step explanation in a Python notebook (.ipynb format) to illustrate how data is processed, split, and used during training, validation, and testing. Additionally, the negative sampling strategy should be described more explicitly, including how and when negative samples are generated. Ensuring that negative samples are drawn only from the training set and not from the test set is essential to avoid data leakage.

---

## Round 0.3 · accepted · Accept

Dear Authors,

Your paper has been revised. It has been accepted for publication in PEERJ Computer Science. Thank you for your fine contribution.

·

Basic reporting

no comment.

Experimental design

no comment.

Validity of the findings

no comment

Additional comments

Thank you to the authors. I believe the necessary corrections have been made.